# Analysis of land cover evolution within the built-up areas of provincial capital cities in northeastern China based on nighttime light data and Landsat data

Zhiwei Xie[☯], Yaohui Han[¤☯], Lishuang Sun[ID]*, Jiwei Ping

School of Transportation Engineering, Shenyang Jianzhu University, Hunnan District, Shenyang, China

☯ These authors contributed equally to this work.
¤ Current address: Liaoning Water Conservancy and Hydropower Survey and Design Research Institute CO., LTD, Heping District, Shenyang, China
* 645695280@qq.com

**Data Availability Statement:** All relevant data in the article can be downloaded at the following URL: https://doi.org/10.5061/dryad.n02v6wwv4.

## Abstract

Mastering the evolution of urban land cover is important for urban management and planning. In this paper, a method for analyzing land cover evolution within urban built-up areas based on nighttime light data and Landsat data is proposed. The method solves the problem of inaccurate descriptions of urban built-up area boundaries from the use of single-source diurnal or nocturnal remote sensing data and was able to achieve an effective analysis of land cover evolution within built-up areas. Four main procedures are involved: (1) The neighborhood extremum method and maximum likelihood method are used to extract nighttime light data and the urban built-up area boundaries from the Landsat data, respectively; (2) multisource urban boundaries are obtained using boundary pixel fusion of the nighttime light data and Landsat urban built-up area boundaries; (3) the maximum likelihood method is used to classify Landsat data within multisource urban boundaries into land cover classes, such as impervious surface, vegetation and water, and to calculate landscape indexes, such as overall landscape trends, degree of fragmentation and degree of aggregation; (4) the changes in the multisource urban boundaries and landscape indexes were obtained using the abovementioned methods, which were supported by multitemporal nighttime light data and Landsat data, to model the urban land cover evolution. Using the cities of Shenyang, Changchun and Harbin in northeastern China as experimental areas, the multitemporal landscape index showed that the integration and aggregation of land cover in the urban areas had an increasing trend, the natural environment of Shenyang and Harbin was improving, while Changchun laid more emphasis on the construction of artificial facilities. At the same time, the method proposed in this paper to extract built-up areas from multi-source city data showed that the user accuracy, production accuracy, overall accuracy and Kappa coefficient are at least 3%, 1%, 1% and 0.04 higher than the single-source data method.

**Funding:** 1) Specific grant numbers: 2. 2) Grant numbers awarded to each author: lnqn201917; L19AJY008. 3) The full name of each funder: Education Department Scientific Research Project of Liaoning Province (CN); Social Science Planning Fund Project of Liaoning Province (CN). 4) Initials of authors who received each award: Zhiwei Xie; Zhiwei Xie. 5) Full names of commercial companies that funded the study or authors: Education Department of Liaoning Province; Social Science Planning Fund Office of Liaoning Province. 6) Initials of authors who received salary or other funding from commercial companies: Nobody. 7) URLs to sponsors' websites: http://jyt.ln.gov.cn/; http://www.lnsgdb.com.cn/Lnsgdb/publish/html/103/index/index.html. 8) The funders had no role in study design, data collection and analysis, decision to publish, or preparation of the manuscript.

**Competing interests:** The authors have declared that no competing interests exist.

## Introduction

Urbanization is a complex phenomenon that has a profound relationship with land cover evolution (LCE) and social dynamics [1, 2]. As the urbanization process has progressed, it has changed from continuous expansion to compression within compact areas. Compact urbanization mainly reflects land cover evolution within urban built-up areas (UBAs), which is manifested by changes in surface landscape patterns, such as impervious ground, vegetation and water [3–5]. LCE can directly reflect the trend of urban master planning in a specific period, which has important practical significance for urban planning and management.

LCE is a description of the changing characteristics and patterns of land surface morphology, not only to analyze the shifts between land cover classes, but also to identify the evolutionary patterns implicit [6]. Land cover change detection and evolution pattern analysis are the main research components of LCE [7]. Along with the development of satellite technology and remote sensing image processing technology, remote sensing has played an important role in the analysis of LCE [8]. The main methods for detecting land cover change are the post-classification comparison method and the direct comparison method [9]. Subject to current automatic image change detection techniques, post-classification comparison is still the dominant technical approach, and supervised classification methods are mostly used [10]. Land cover evolution models are used to describe, interpret, predict, optimize, and support decision making on land cover change. The land cover evolution model analysis model mainly includes quantitative analysis model, ecological quality analysis model, land use degree model, spatial transfer change analysis model and change prediction model [11, 12]. Most of the above models use a single or composite index to express land change indicators and pay less attention to the evolution of the spatial distribution pattern.

The urban landscape pattern has a strong correlation with the spatial distribution pattern of impervious ground, vegetation and water, which can be applied to the study of the urban heat island effect, green space optimization, the mechanism of water logging disaster formation, and other issues [13, 14]. For the study of compact urbanization, researchers have analyzed the trends of urban construction by analyzing changes in landscape patterns within built-up areas to quantitatively reflect LCE [15, 16]. The landscape index provides morphological information about highly condensed landscapes that reflects the structural composition and spatial configuration of the landscape elements. Fragstats software is one of the landscape index analysis platforms that is widely used by researchers [17, 18].

Through sprawling urbanization, the rate of urbanization in the post-World War II developing world has been remarkable. However, the sprawling urbanization has resulted in the decay of the original urban areas, and the central urban area at night has become an empty city [19]. This has resulted in a huge waste of land and energy resources, and has caused ecological damage around the city. As a result, there is a trend towards developing compact urbanization and an urgent need to assess the land cover evolution within built-up [20] areas. Using remote sensing image data to determine urban built-up area boundaries (UBAB) is an important process for analysis of LCE and compact urbanization. Depending on the number of data sources used, the methods for extracting UBA can be divided into single-source data and multisource data methods. The single-source data method generally uses nighttime light data or Landsat data separately. In recent years, due to the strong stability of nighttime light data and the high integrity of UBA extraction results, many studies have been conducted, such as those on the neighborhood extreme value method, statistical data reference method and gradient spatial constraint method [21–23]. However, nighttime light data can only capture the spatial extent of urban areas with obvious light at night, which does not include factories, development zones and other areas at the edges of cities that are not lit at night [24, 25]. Landsat data are

also widely used to extract UBA, but the complexity of spectral features has resulted in the misclassification of some land cover types [26]. In addition, the completeness of the extraction results is insufficient, and the gap areas formed by the vegetation and water need to be filled manually [27]. The combination of nighttime light data and Landsat data can improve the accuracy of UBA extraction, and researchers have proposed methods, such as the fusion variable method, weighted pixel averaging method, and feature fusion method [28–30].

China is the second-largest economy in the world and is urbanizing at a remarkable speed. Northeastern China, which is an old industrial base, plays an important role in China's economic structure. Since the implementation of the policy of revitalizing the old industrial base in northeastern China in 2003, the urbanization rate in this area has been the highest in the country. Due to the rapid expansion of urban areas, the quality of urbanization in northeastern China is poor [31, 32]. To develop a compact urbanization, it is of great practical significance to ascertain the landscape pattern changes in the capital cities of northeastern China and to gain insight into the LCE [33].

In this paper, we propose a method for analyzing the land cover evolution within built-up areas that can obtain the tendency of urban land construction and allow a deeper understanding of compact urbanization. First, the neighborhood extreme method and maximum likelihood method are used to extract the UBAB. Second, the single-source UBAB data described above is integrated with the weighted boundary pixel fusion algorithm to build the multisource UBAB. Then, Landsat data within the UBAB are classified into classes, such as impervious ground, vegetation and water, by the maximum likelihood method, and the corresponding landscape pattern indexes are calculated. Finally, the multitemporal landscape pattern indexes are used to obtain the landscape pattern changes and to analyze the land cover evolution trends. To verify the effectiveness of the method, Shenyang, Changchun and Harbin, capital cities in northeastern China, were used as experimental areas and compared with other methods. At the end of this paper, the main advantages and disadvantages of this method are described. The technical flowchart of this paper is shown in Fig 1.

## Materials and methods

### Study areas

This study selected three capital cities in northeastern China as the research areas: Shenyang city, Liaoning Province; Changchun city, Jilin Province; and Harbin city, Heilongjiang Province. As China's first heavy industrial and agricultural bases, the capital cities of these three northeastern provinces had earlier economic development and once had the highest level of urbanization in China. In addition, these cities are the political, economic, cultural, educational and transportation centers of the region. Shenyang is the largest city in northeastern China, with a total area of 13,000 square kilometers and a resident population of 8,316,000 in 2019 [34]; Changchun is located in the geographical center of northeastern China, with a total area of 20,500 square kilometers and a resident population of 7,667,000 in 2019 [35]; and Harbin is the northernmost provincial capital city in China, with a total area of 53,100 square kilometers and a resident population of 9,515,000 in 2019 [36].

### Data

**Nighttime light data.**   The nighttime light data used in this paper were the DMSP/OLS (Defense Meteorological Satellite Program/Operational Linescan System) published by NOAA. These data have the advantages of long time series and high stability and are widely used to extract urban built-up areas [37–39]. The DMSP/OLS data range from 1992 to 2013 and have a spatial resolution of 1000 m. In this study, the DMSP/OLS stable nighttime light

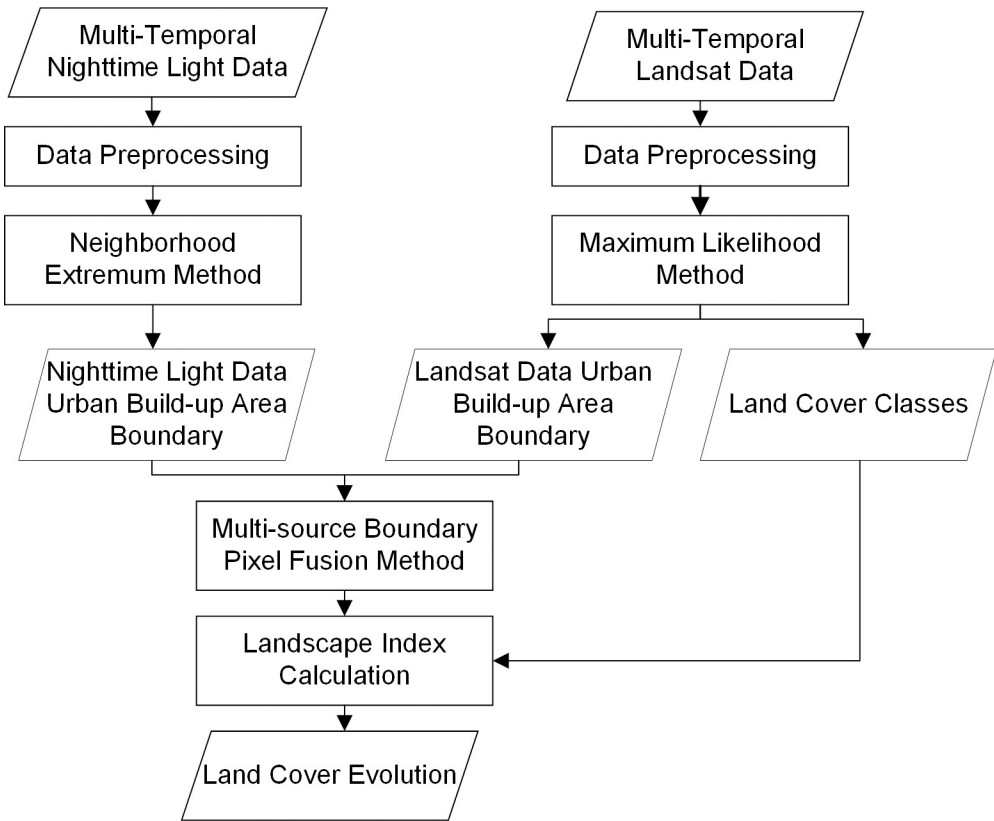

**Fig 1. Analysis of land cover evolution within built-up areas based on nighttime light data and Landsat data.**

data from 2000 to 2012 were selected, and the effects of oversaturation and discontinuity were reduced by relative radiometric correction.

**Landsat data.** Landsat imagery courtesy of NASA Goddard Space Flight Center and U.S. Geological Survey, downloaded from USGS (https://earthexplorer.usgs.gov/). Landsat data are image data published by NASA and are widely used to analyze urbanization due to their high spatial resolution. In this paper, Landsat TM data from 2000 to 2012 were selected for the extraction of built-up areas; data with interference from the sensors or the atmosphere were excluded. In addition, the reference built-up areas that were consistent with the experimental time period were identified by artificial interpretation [40].

## Methodology

### Data preprocessing

**Nighttime light data.** The DMSP/OLS data were set to a Lambert isometric cone projection with a resampling accuracy of 1 km. To address the grayscale saturation of DMSP/OLS, we selected the 2017 F16 data as the baseline data and applied the following one-dimensional quadratic regression model to the other data for saturation correction [39]:

$$DN_b = \begin{cases} p * DN^2 + q * DN + m, DN_c, DN_c < 63 \\ 63, DN_c \geq 63 \end{cases}, \tag{1}$$

where $DN$ and $DN_b$ represent the grayscale values of the image before and after correction, respectively; $p$ and $q$ are regression coefficients, and $m$ is a constant.

To address the problem of fluctuating gray values in multitemporal nighttime light data, formula (2) was used to perform a relative radiation correction for the image.

$$sDN_{(n,i)} = \begin{cases} DN_{(n-1,i)}, DN_{(n-1,i)} > DN_{(n,i)} \\ DN_{(n,i)}, otherwise \end{cases}, \tag{2}$$

where $DN_{(n-1,i)}$ and $DN_{(n,i)}$ represent the gray values of pixel $i$ in years $n-1$ and $n$, respectively.

**Landsat data.** Landsat imagery courtesy of NASA Goddard Space Flight Center and U.S. Geological Survey, downloaded from USGS (http://landsat.visibleearth.nasa.gov/). To eliminate the effects of atmospheric water vapor, oxygen and methane on the radiation from the features [41], the FLAASH model was used for atmospheric correction of the images. Subsequently, the multitemporal remote sensing images were calibrated using a polynomial algorithm.

## Urban built-up area extraction based on single source data

**Urban built-up area extraction based on the neighborhood extremum method using DMSP/OLS data.** The neighborhood extremum method uses the change in grayscale of the nighttime light data from the built-up to the non-built-up areas as the basis, using the locations of gray mutation point as boundary points. The closed curve formed by combining all boundary points is the UBAB. First, the linear filtering method is used to construct the neighborhood difference image. The gray values of the DMSP/OLS data gradually decrease from built-up to non-built-up areas and show a gradient of changes. To enhance the grayscale gradient variation, an 8-neighborhood extremum template was constructed. The template calculates the absolute difference in the gray values between the central image and the rest of the images; the maximum difference is used as the grayscale value of the central image. The template calculates the absolute difference in the gray values between the central pixel and the rest of the pixels, taking the maximum difference as the gray value of the central pixel. The entire image is filtered using the 8-neighborhood extreme template, and a neighborhood difference image reflecting the grayscale variation feature was obtained, as shown in Fig 2. The black and white areas in the figure correspond to areas with small and large changes in gray values, respectively.

Then, the UBABs were extracted using an extreme value search method. The neighborhood difference pixels at the junction of the built-up and non-built-up areas have the greatest

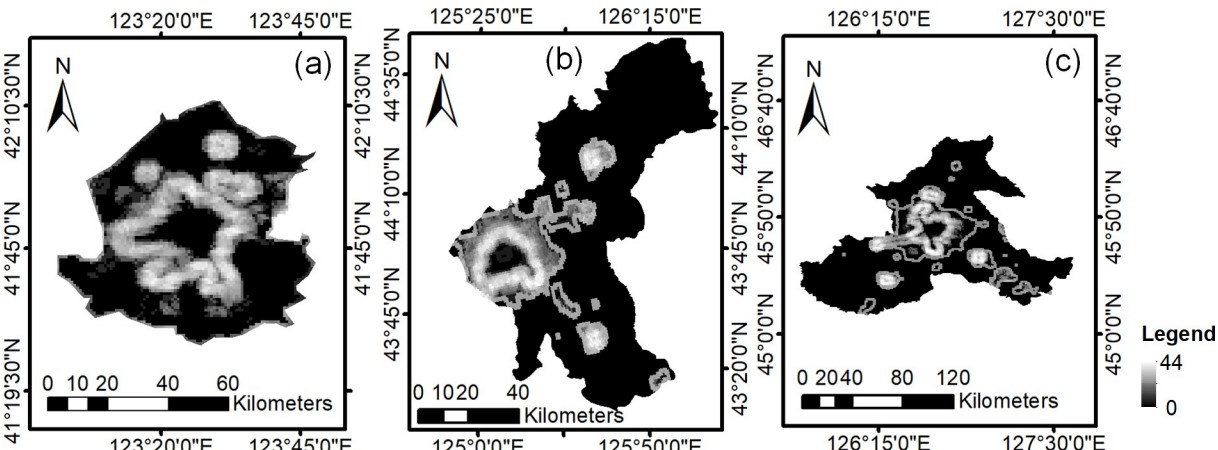

**Fig 2. Neighborhood difference image.** (a)Shenyang;(b) Changchun;(c) Harbin.

variation in gray values, which can be expressed as follows:

$$DN_{(i,j-1)} < DN_{(i,j)} < DN_{(i,j+1)}, \tag{3}$$

where $DN_{(i,j)}$ is the target pixel and represents the gray value of the pixel in row $i$, column $j$. We extracted the pixels that satisfy the above formula, set the remaining pixels to 0, and constructed the boundary extreme image, as shown in Fig 3(A), 3(B) and 3(C). The gray values of the cross section of boundary extreme images are extracted, and they are shown in Fig 3(A), 3(E) and 3(F). The collection of pixels with gray values greater than or equal to the lowest peak point was used as the boundary, resulting in the boundary image. Finally, object-oriented classification was used to extract the built-up areas of the boundary image, and the UBAB extraction from the DMSP/OLS data was achieved.

**Urban built-up area extraction based on the maximum likelihood method using landsat data.** Remote sensing image classification methods are divided into supervised classification and unsupervised classification. Unsupervised classification only depends on the distribution of pixel gray value in spectral feature space, which makes it difficult to achieve accurate matching between pixel clusters and ground classes. Because of its simple calculation and strong stability, maximum likelihood method is still widely used in supervised classification algorithm [42]. The maximum likelihood method is an image classification method that combines the distances of samples to the centers of known classes and spectral distribution features. This method is widely used to extract urban built-up area from Landsat data [43]. The image classes were impervious surface, vegetation and water, and 100 training samples were selected for each category through artificial interpretation. The mean vector and covariance matrix of the training samples were calculated to obtain the corresponding probability discriminant

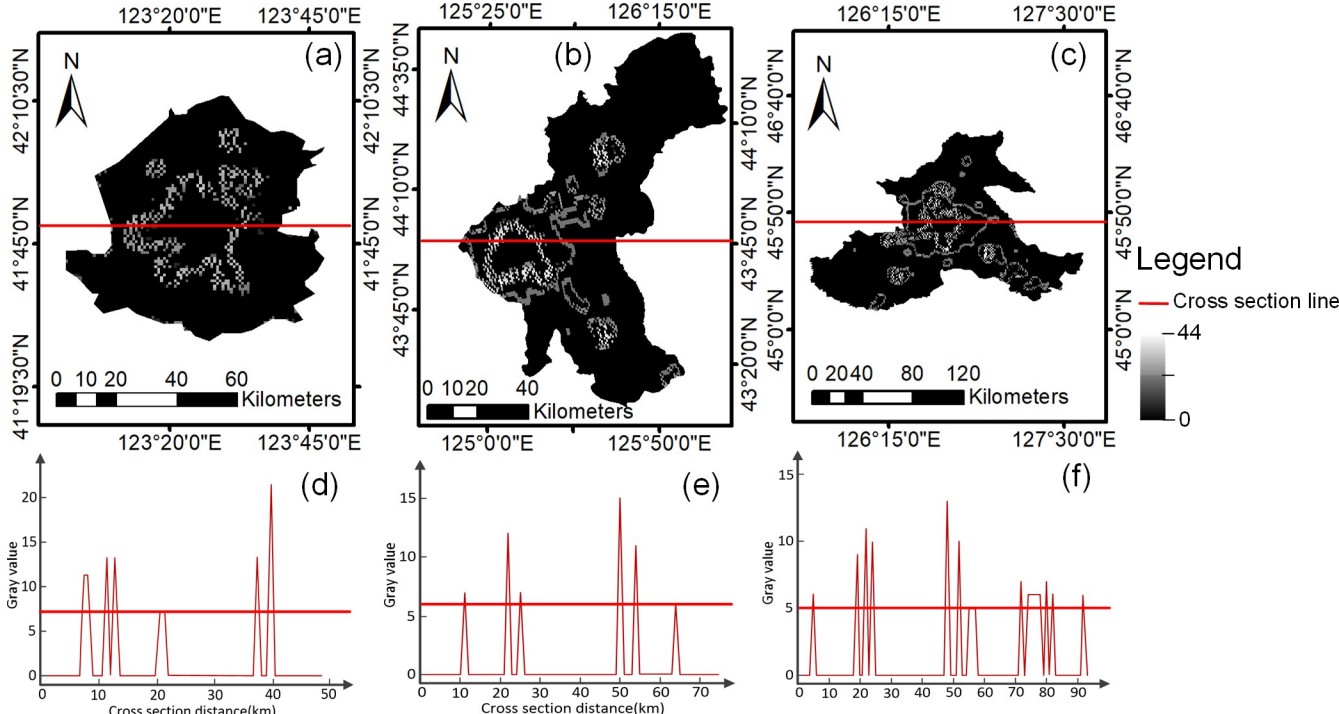

**Fig 3. Boundary extreme images and diagram of cross section line gray values.** (a) Boundary extreme images of Shenyang; (b) Boundary extreme images of Changchun; (c) Boundary extreme images of Harbin; (d) Schematic of cross section line gray values of Shenyang; (e) Schematic of cross section line gray values of Changchun; (f) Schematic of cross section line gray values of Harbin.

function and Bayesian discriminant rule. Finally, the image data were classified, and the classification results were postprocessed with clumping, sieving, removal of outliers, etc. The pixels belonging to the impervious surface class were used as built-up areas, while the water and vegetation (urban green spaces) located within the construction land were also used as built-up areas and the rest as non-built-up areas, thus enabling the UBAB extraction.

## Urban built-up areas extraction based on the multisource boundary pixels fusion method

**Advantages and disadvantages of urban built-up area extraction using single-source data.** The results of the urban built-up area boundary extraction with DMSP/OLS data and Landsat data are shown in Fig 4, and the advantages and disadvantages of the above results were assessed as follows:

1. The extraction of the urban built-up area boundaries using DMSP/OLS data had the advantage of stability and high integrity. The light sensitivity of the DMSP/OLS data may be missed in areas with weak lighting, as shown in Fig 4(B). In addition, the light spillover nature of DMSP/OLS data has led to a wide range of UBABs.

2. The high spatial resolution of the Landsat data makes it possible to better reflect details. Landsat data have difficulty obtaining high-quality time series due to clouds and sensor errors. The spectral complexity of the data also affects the completeness and correctness of the results, as shown in Fig 4(A).

Based on the above analysis, to solve the problem of omission detection in built-up areas extracted from single source data, this paper proposes a multisource boundary pixel fusion method. The method combines the advantages of multisource data to optimize UBAB.

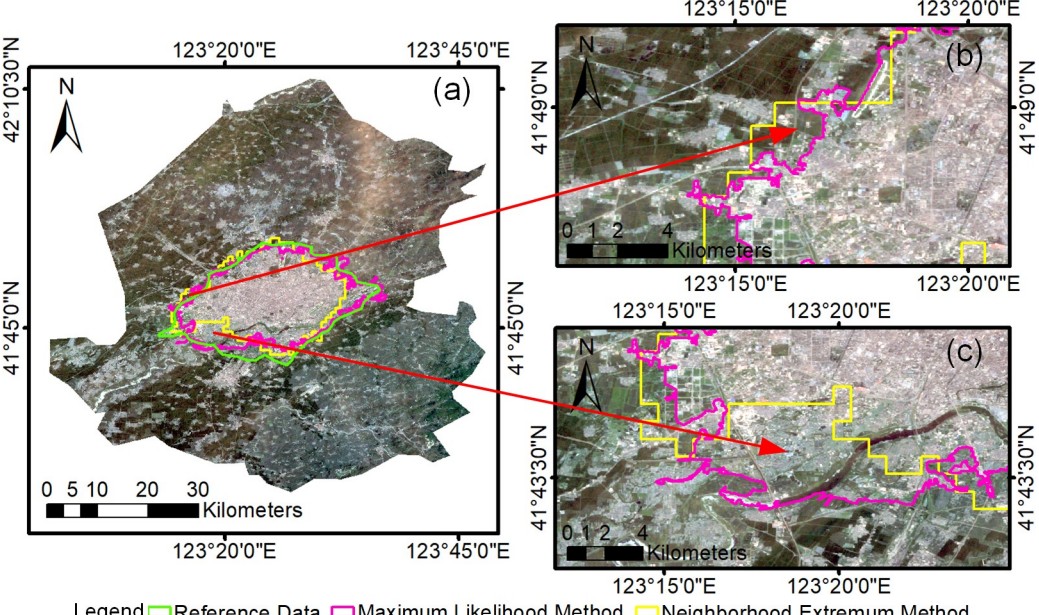

**Fig 4. Advantages and disadvantages of the urban built-up area boundary extraction based on single source data.** (a) Built-up area extraction results of Shenyang in 2000 based on single source data; (b) Schematic of omissions in the extraction results with Landsat data; (c) Schematic of omissions in the extraction results with the DMSP/OLS data. (USGS/NASA Landsat).

*Multisource boundary pixel fusion method*. Landsat data and DMSP/OLS built-up area boundaries were used as reference boundaries and target boundaries, as shown in Fig 5.

First, the tangent L is constructed for each pixel $O(x_0,y_0)$ in the reference boundary. Next, the normal distribution of point $O$ is constructed; the intersection with the target boundary is points $A$ and $B$, and the nearest intersection $A(x_1,y_1)$ is chosen. Finally, the fusion of points $O$ and $A$ is calculated using Eq (4) to obtain the fusion point $C(x,y)$. Multisource data fusion boundaries are formed by the fusion points as follows:

$$\begin{cases} x = ax_0 + bx_1 \\ y = ay_0 + by_1 \end{cases}, \qquad (4)$$

where $a$ and $b$ are the fusion weights, $a+b = 1$ and in general $a = b = 0.5$.

## Analysis of land cover evolution within built-up areas based on landscape index

The urban built-up area boundaries were obtained using the multisource boundary pixel fusion method, and land cover classes, such as impervious surface, water and vegetation, were obtained by classifying the Landsat data within the boundaries by the maximum likelihood method. We calculated the landscape index to reflect land cover evolution. The principle of selecting landscape indexes is that they can represent the characteristics of ecological state and landscape pattern, and have low redundancy [44]. Landscape index can be divided into three categories: overall trend index, fragmentation index and aggregation index [45]. In combination with the research focus of compact urbanization in this paper, Class area (CA), Percent of landscape (PL), Patch density (PD), Aggregation index (AI) were selected from above categories respectively.

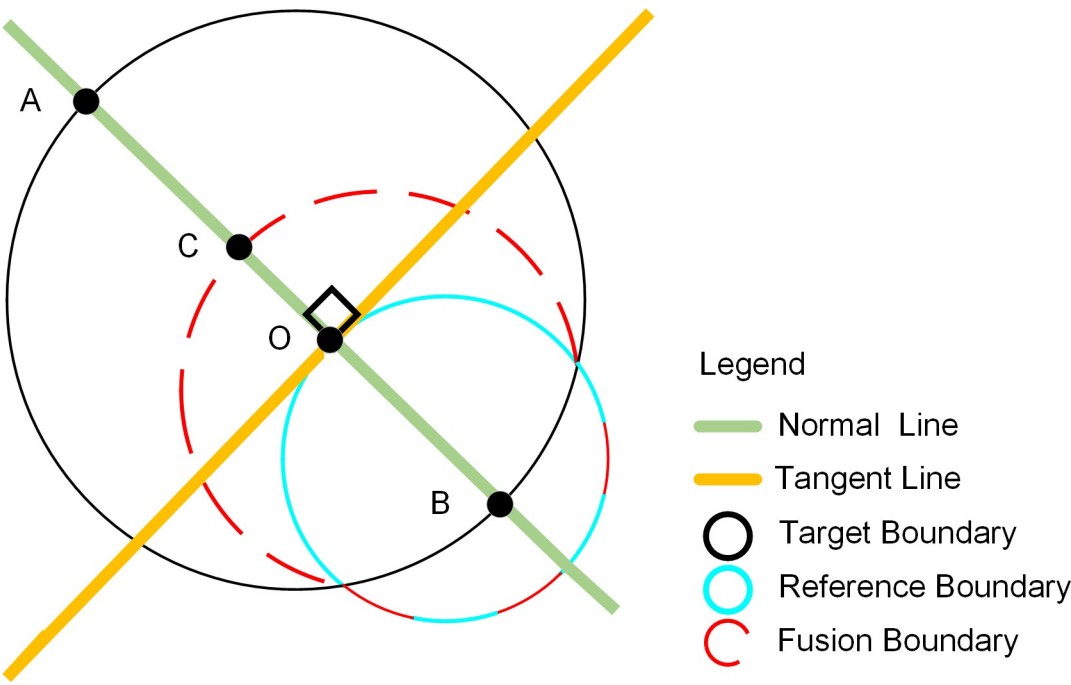

**Fig 5. Schematic of the multisource data boundary fusion method.**

(1) Overall trend index

① Class area (CA): CA is the size of the land cover and reflects the overall level of development of the land cover within UBA; it can be expressed as follows:

$$CA = a_i, \tag{5}$$

where $a_i$ is the area of class $i$.

② Percent of landscape (PL): PL is a measure of the landscape composition that can be used to assess the tendency of land cover change within an UBA. The formula is as follows:

$$PLAND = \frac{\sum_{j=1}^{n} a_{ij}}{A}, \tag{6}$$

where $a_{ij}$ is the area of patch $j$ that belongs to class $i$. $A$ denotes the total landscape area and $0 < PLAND < 1$.

(2) Fragmentation index

Patch density (PD): PD is the class number of land cover in a unit area, judging the fragmentation degree of land cover within UBA. The formula is as follows:

$$PD = \frac{b_n}{A}, \tag{7}$$

where $b_n$ is the number of patch $n$, and $A$ is the total area of all patches.

(3) Aggregation index

Aggregation index (AI): AI reflects the aggregation degree of the patches belonging to the same class and the spatial configuration of the land cover. The formula is as follows:

$$AI = (\frac{g_i}{\max \to g_i}) \times 100, \tag{8}$$

where $g_i$ is the number of adjacent patches belonging to class $i$ and $\max \to g_i$ is the largest number of adjacent patches.

## Results

### Urban built-up area extraction

The DMSP/OLS data and Landsat data from 2000, 2004, 2008 and 2012 were used as experimental data. The extraction results of neighborhood extremum method and maximum likelihood method are shown in Figs 6 and 7.

The UBABs of Shenyang, Changchun and Harbin were extracted with the multisource boundary pixel fusion method, and the results are shown in Fig 8 and Table 1. The results showed that the UBA in all three cities had shown a clear growth trend. Shenyang, Changchun and Harbin had the fastest growth in the urban built-up areas from 2008 to 2012, 2008 to 2012 and 2000 to 2004, respectively.

### Analysis of land cover evolution within urban built-up areas

**Land cover classification.**   The Landsat data were clipped using the urban built-up area boundaries and classified into impervious surfaces, water and vegetation using the maximum likelihood method, and the classification results are shown in Fig 7 and Table 2. As shown in Fig 9, the land cover classes within the three UBAs showed trends of expansion, with a higher degree of expansion in impervious surfaces and vegetation. The average growth rates of the

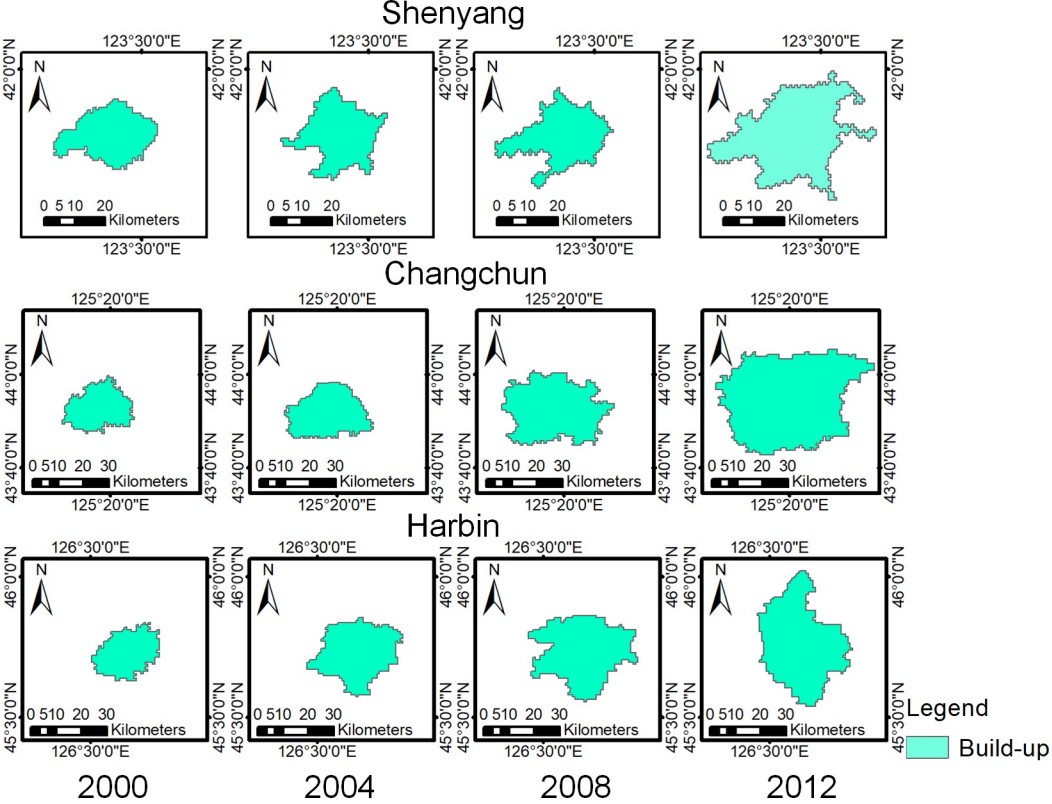

**Fig 6. Extraction results of the urban built-up areas by the neighborhood extremum method.**

Shenyang land cover classes were above 20%, with the highest growth in vegetation from 2000 to 2004 and in water and impervious surfaces from 2004 to 2008. The average growth rate of the impervious surfaces in Changchun was approximately 50%, with the fastest growth period from 2000 to 2004. The growth rate of land cover in Harbin was approximately 30%, with the fastest growth in vegetation and impervious surfaces occurring from 2000 to 2004 and in water from 2004 to 2008.

**Land cover evolution analysis based on landscape pattern index.** The landscape indexes of the impervious surfaces, vegetation and water were calculated separately, including the CA, PL, PD and AI, and the statistical results are shown in Table 3. The specific analysis was as follows:

1. Shenyang: Landscape index trend chart for Shenyang is shown in Fig 10. CA: all land cover classes showed an increasing trend, with the fastest growth in impervious surfaces, which reflected the expansion of the built-up areas; PL: the impervious surfaces covered the largest percentage of the built-up areas and showed a decreasing trend, while vegetation and water occupancy showed increasing trends; PD: all land cover classes showed downward trends, with the largest rate of decline in vegetation, which reflected reduced fragmentation and increased homogeneity of the land cover; AI: all land cover classes showed downward trends, with the highest rate of decline in vegetation, which reflected the increased agglomeration of land cover classes. In general, impervious surface indexes had the highest values, but the occupancy, aggregation and integrity of vegetation increased, and the ecological environment of Shenyang tended to be optimized.

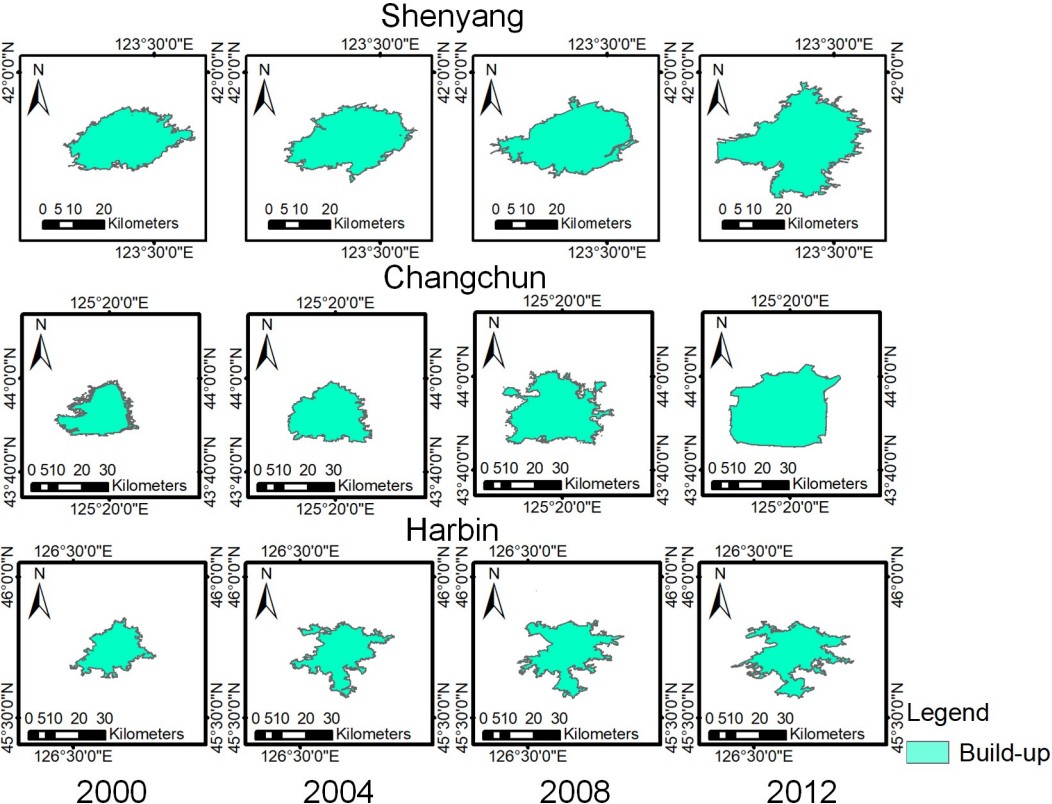

**Fig 7. Extraction results of the urban built-up areas by the maximum likelihood method.** (USGS/NASA Landsat).

2. Changchun: Landscape index trend chart for Shenyang is shown in Fig 11. CA: all land cover classes showed increasing trends, with the highest rate of growth in impervious surfaces, which reflected the expansion of built-up areas. PL: the impervious surface showed an upward trend, while vegetation and water showed downward trends, reflecting the fact that the area occupied by impervious surfaces was still increasing; PD: all land cover classes showed a downward trend, with the highest rate of decline in vegetation, reflecting the reduced fragmentation of the land cover; AI: the impervious surface and vegetation show a downward trend and the water showed an upward trend, reflecting increasing aggregation in the former classes and decreasing aggregation in the latter. Overall, the impervious surface of Changchun is being further strengthened, with greater emphasis on the construction of artificial features.

3. Harbin: Landscape index trend chart for Shenyang is shown in Fig 12. CA: all land cover classes showed growing trends, with impervious surfaces showing the highest growth rate, which reflected their greater contribution to urban expansion. PL: the proportion of impervious surface was the highest, but the proportion of vegetation was increasing, reflecting the improving natural environment of the city; PD: the fragmentation of all land cover classes showed a decreasing trend, with vegetation fragmentation decreasing the fastest, reflecting the increasing homogeneity of land cover; AI: the areas of all land cover classes increased, but the trend towards impervious surfaces was more pronounced. Overall, ecological construction in Harbin was increasing, while the land cover was in an aggregated state.

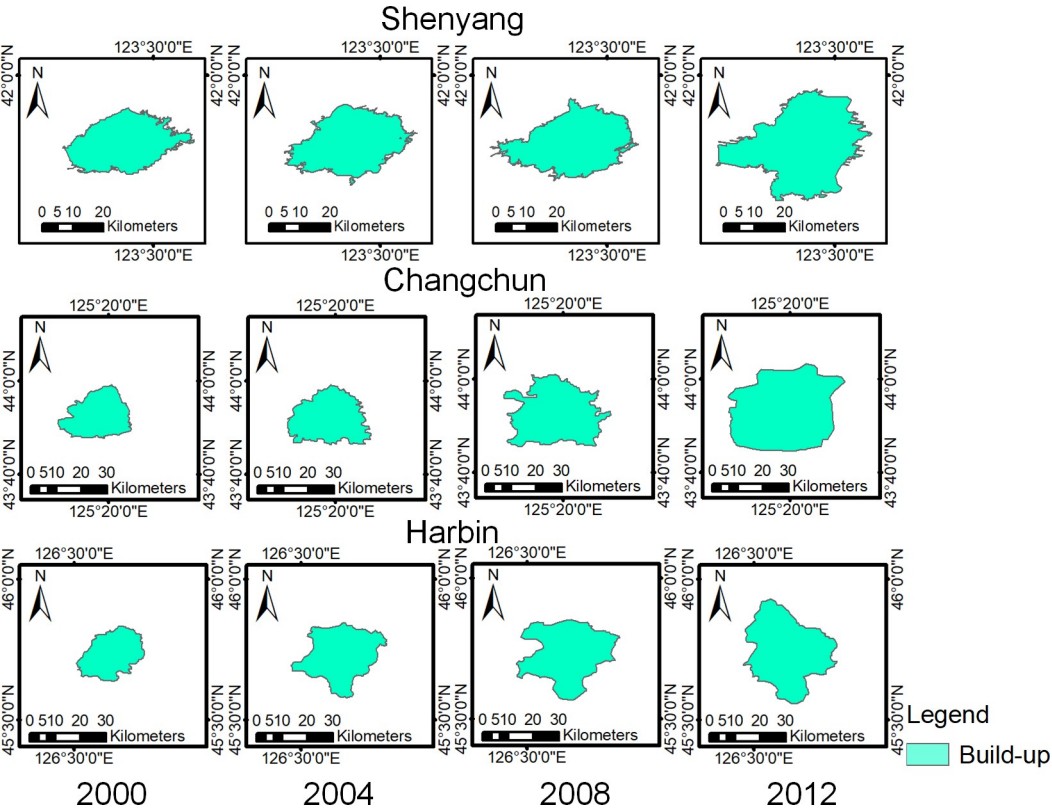

**Fig 8. Extraction results of the urban built-up areas by the multisource boundary pixel fusion method.** (USGS/NASA Landsat).

In summary, the spatial extent of the land cover classes in Shenyang, Changchun and Harbin all showed growing trends, with increases in both integration and aggregation. In addition, the areas occupied by vegetation in Shenyang and Harbin were increasing, indicating that the natural environment was improving, while Changchun was focused on the construction of artificial facilities.

## Discussion

### Urban built-up area extraction accuracy

To verify the validity of this paper's method, the built-up areas of Shenyang, Changchun and Harbin were extracted using the artificial interpretation method. The built-up areas of the DMSP/OLS and Landsat data were extracted with the neighborhood extreme value method and the maximum likelihood method, respectively, and compared with the multisource boundary pixel fusion method.

**Table 1. Statistics of urban built-up areas (km$^2$).**

|  | 2000 | 2004 | 2008 | 2012 | 2000–2004 | 2004–2008 | 2008–2012 | MAX |
|---|---|---|---|---|---|---|---|---|
| **Shenyang** | 408 | 464 | 495 | 740 | 13.70% | 6.60% | 49.40% | 2008–2012 |
| **Changchun** | 288 | 382 | 555 | 848 | 32.60% | 45.20% | 52.70% | 2008–2012 |
| **Harbin** | 266 | 427 | 530 | 619 | 60.50% | 24.10% | 16.70% | 2000–2004 |

**Table 2. Statistics of land cover classification from 2000 to 2012 (km$^2$).**

| | Year | 2000 | 2004 | 2008 | 2012 | 2000–2004 | 2004–2008 | 2008–2012 | Average | Max |
|---|---|---|---|---|---|---|---|---|---|---|
| | V | 112 | 161 | 186 | 198 | 43.70% | 15.50% | 6.40% | 21.87% | 2000–2004 |
| S | W | 10 | 12 | 20 | 21 | 20.00% | 66.60% | 0.50% | 29.09% | 2004–2008 |
| | I | 271 | 310 | 391 | 470 | 14.30% | 26.10% | 20.20% | 20.20% | 2004–2008 |
| | V | 113 | 110 | 197 | 295 | -2.60% | 79.00% | 49.70% | 40.03% | 2004–2008 |
| C | W | 10 | 11 | 12 | 21 | 10.00% | 9.00% | 75.00% | 31.33% | 2008–2012 |
| | I | 161 | 259 | 344 | 530 | 60.80% | 32.80% | 54.00% | 49.20% | 2000–2004 |
| | V | 94 | 154 | 212 | 214 | 63.80% | 37.00% | 0.90% | 33.90% | 2000–2004 |
| H | W | 9 | 12 | 17 | 23 | 33.30% | 41.60% | 35.20% | 36.70% | 2004–2008 |
| | I | 161 | 259 | 298 | 320 | 60.80% | 15.00% | 7.30% | 27.70% | 2000–2004 |

S, Shenyang; Changchun; H, Harbin; V, Vegetation; W, Water; I, Impervious Surface.

As seen from Table 4 the user accuracy, production accuracy, overall accuracy and kappa coefficients of the boundary image fusion method were 88%, 90%, 99% and 0.88, respectively, which were at least 3%, 1%, 1% and 0.04 better than the other two methods. The method in this paper had higher accuracy than the built-up area extraction method based on single source data.

## Land cover classification accuracy

To assess the accuracy of the land cover classification, the classification results of water, vegetation and impervious surfaces were assessed for accuracy. The 100 validation samples were

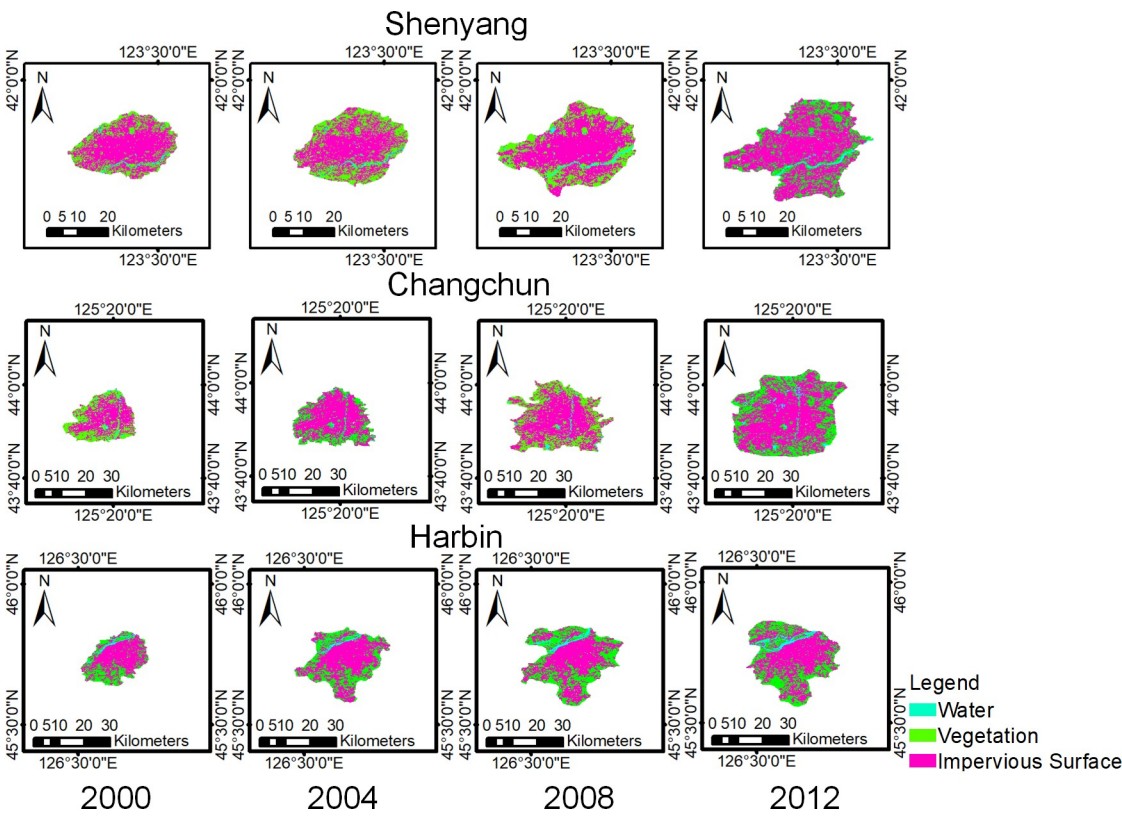

**Fig 9. Land cover class results.** (USGS/NASA Landsat).

**Table 3. Landscape index within urban build-up areas from 2000 to 2012.**

| Y | C | Shenyang | | | | Changchun | | | | Harbin | | | |
|---|---|---|---|---|---|---|---|---|---|---|---|---|---|
| | | CA | PL | PD | AI | CA | PL | PD | AI | CA | PL | PD | AI |
| **2000** | I | 27089.64 | 0.69 | 0.63 | 85.41 | 15509.79 | 0.54 | 1.12 | 82.02 | 16241.51 | 0.61 | 0.99 | 85.60 |
| | V | 11331.90 | 0.29 | 2.25 | 64.19 | 11806.10 | 0.41 | 2.13 | 74.80 | 9403.49 | 0.35 | 2.17 | 74.44 |
| | W | 1047.33 | 0.03 | 0.38 | 63.21 | 1192.46 | 0.04 | 1.75 | 43.04 | 947.93 | 0.04 | 0.12 | 77.77 |
| **2004** | I | 28956.00 | 0.60 | 0.92 | 81.19 | 25879.19 | 0.68 | 0.54 | 87.82 | 25928.64 | 0.61 | 0.57 | 85.35 |
| | V | 18015.00 | 0.37 | 1.86 | 69.74 | 11094.43 | 0.29 | 1.53 | 69.38 | 15528.96 | 0.36 | 0.83 | 74.26 |
| | W | 1374.00 | 0.03 | 0.37 | 62.44 | 1159.71 | 0.03 | 1.15 | 37.03 | 1249.92 | 0.03 | 0.31 | 58.14 |
| **2008** | I | 39120.12 | 0.65 | 0.38 | 87.17 | 28828.80 | 0.52 | 0.69 | 79.21 | 29808.22 | 0.56 | 0.53 | 83.50 |
| | V | 18576.48 | 0.31 | 0.69 | 71.32 | 25053.12 | 0.45 | 0.92 | 73.63 | 21321.04 | 0.40 | 0.69 | 75.36 |
| | W | 2077.01 | 0.03 | 0.55 | 50.19 | 1954.08 | 0.04 | 0.55 | 48.41 | 1747.46 | 0.03 | 0.10 | 73.54 |
| **2012** | I | 44662.52 | 0.65 | 0.34 | 80.93 | 52717.86 | 0.62 | 0.40 | 83.53 | 32146.56 | 0.57 | 0.60 | 82.15 |
| | V | 22342.04 | 0.32 | 1.31 | 60.64 | 29852.16 | 0.35 | 0.89 | 70.59 | 21458.88 | 0.38 | 1.06 | 72.81 |
| | W | 1926.68 | 0.03 | 0.27 | 57.17 | 2240.94 | 0.03 | 0.40 | 46.06 | 2381.76 | 0.04 | 0.16 | 72.54 |

Y, Year; C, Class; V, Vegetation; W, Water; I, Impervious Surface.

randomly generated for each class, and their classes were interpreted visually. The interpretation results were compared with the computer classification results, and the average accuracy is shown in Table 5. As shown in Table 5, the overall accuracy and Kappa coefficient of the

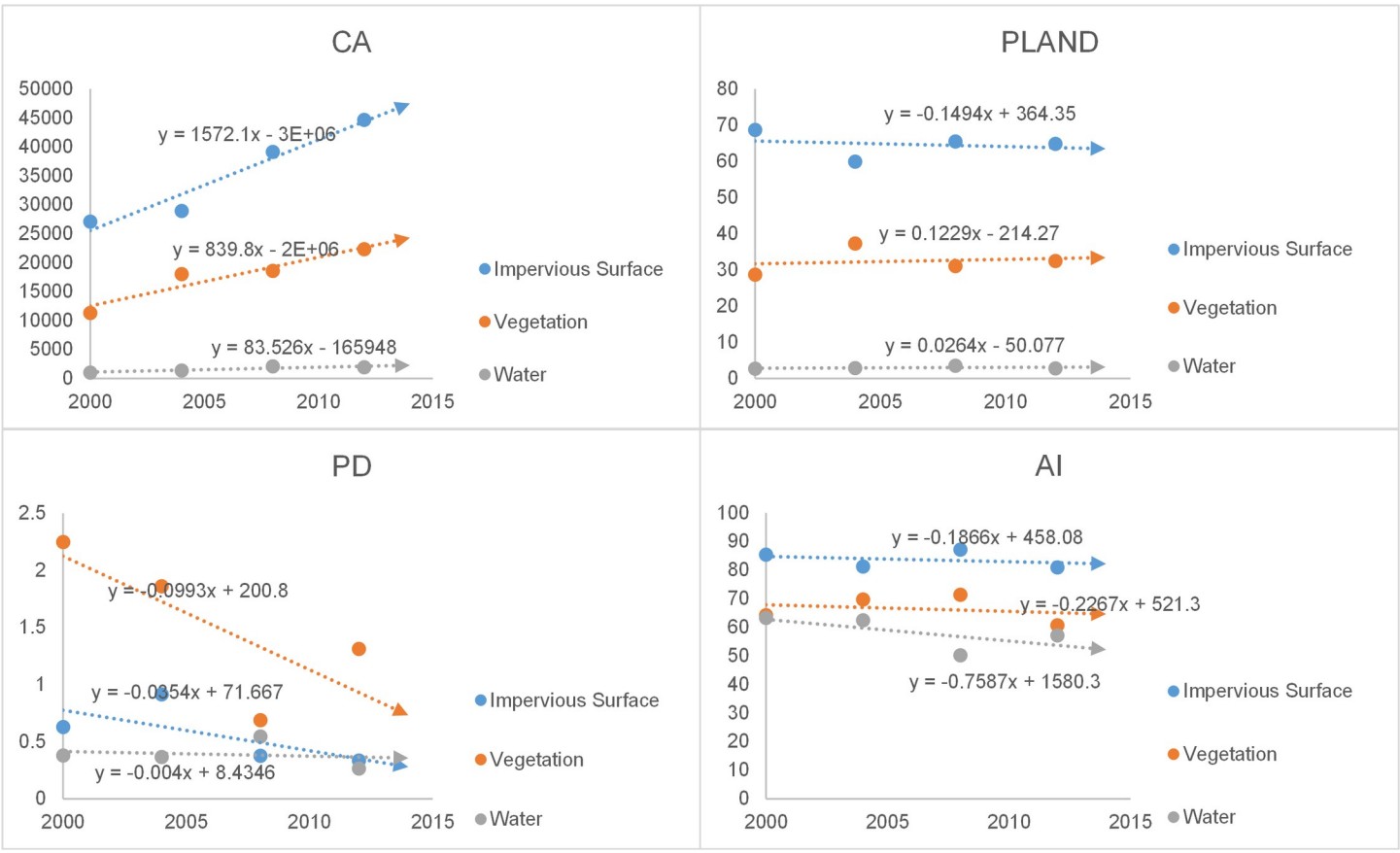

**Fig 10. Landscape index trend chart for Shenyang.**

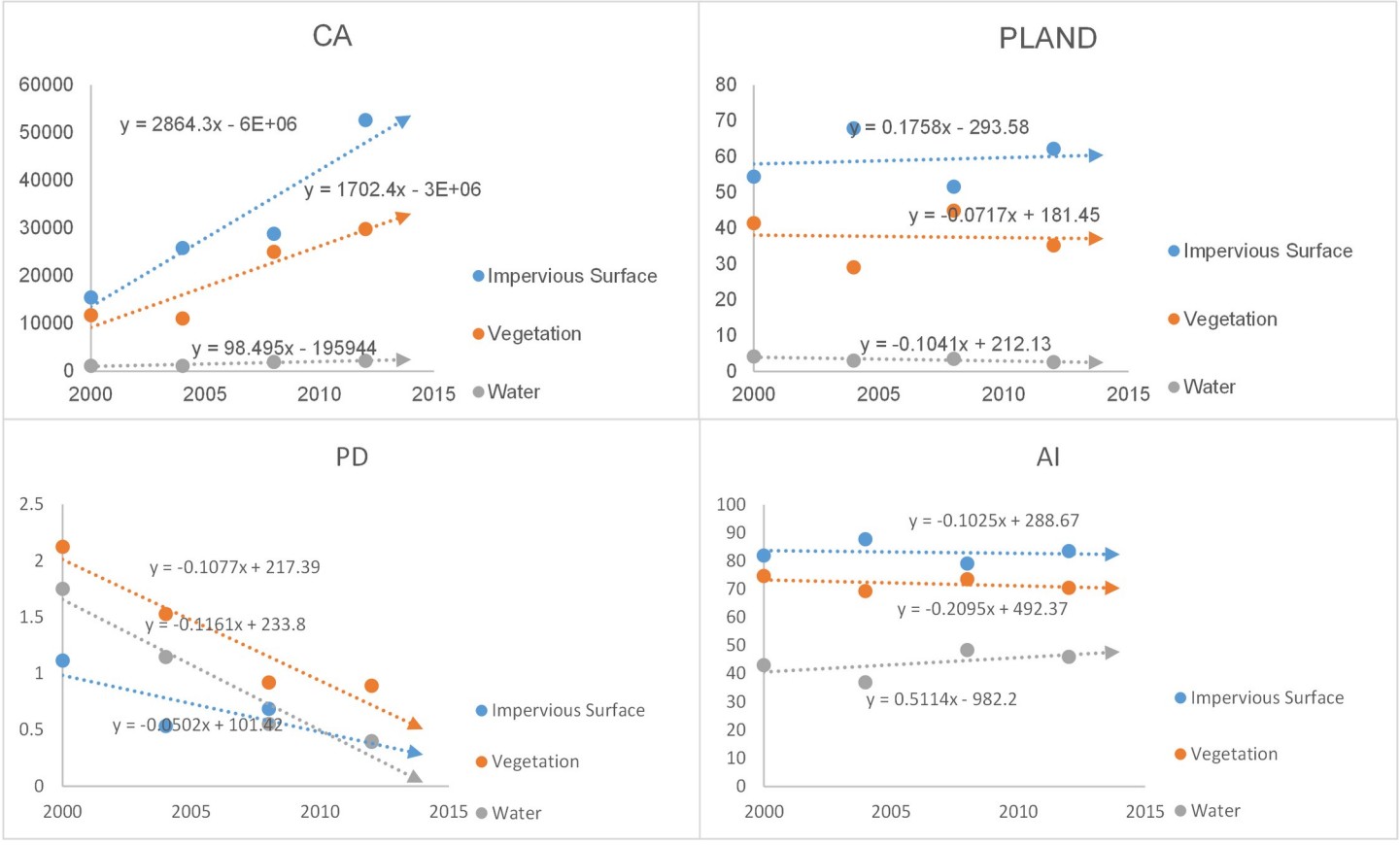

**Fig 11. Landscape index trend chart for Changchun.**

method used in this paper were above 95% and 0.90, respectively, which meets the accuracy requirements.

## Boundary pixel fusion weight analysis

Different combinations of weights for the multisource boundary pixel fusion algorithm can affect the accuracy of built-up area extraction. We set *a* and *b* to different values and calculated the Kappa coefficients of the extraction results, which are shown in Fig 13. As seen from Fig 13, setting the weights to *a* = *b* = 0.5 give the highest accuracy.

## Conclusions

Through analyses of provincial capitals in northeastern China, the validity of the proposed method for analyzing land cover evolution within urban built-up areas was demonstrated. We extracted urban built-up areas using DMSP/OLS data and Landsat data and analyzed land cover evolution within built-up areas using landscape indexes. To improve the extraction accuracy of the built-up areas based on single-source remote sensing data, the multisource boundary pixel fusion method was proposed, which effectively integrates the information advantages of multisource data. Using the urban built-up areas as the spatial extent, the land cover evolutions of Shenyang, Changchun and Harbin from 2000 to 2012 were analyzed through the landscape indexes, and the following results were obtained: the built-up areas exhibited clear expansion trends, with increased integration and agglomeration of land cover classes, such as

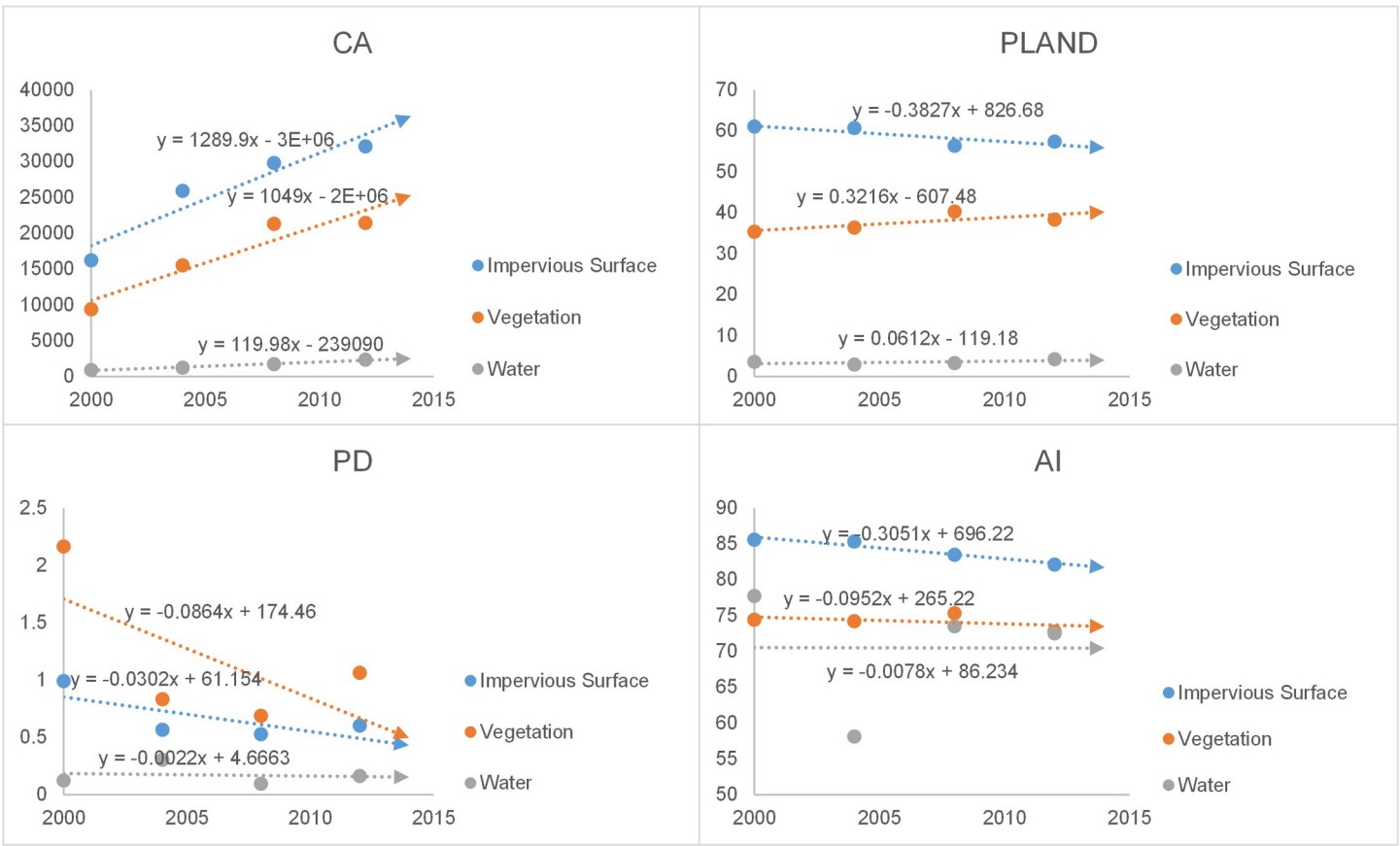

**Fig 12. Landscape index trend chart for Harbin.**

**Table 4. Confusion matrix for the extraction of built-up areas.**

| C | Y | Neighborhood extremum method | | | | | Maximum likelihood method | | | | | Multi-source Boundary pixel fusion method | | | | |
|---|---|------|------|------|------|------|------|------|------|------|------|------|------|------|------|------|
| | | 2000 | 2004 | 2008 | 2012 | Me | 2000 | 2004 | 2008 | 2012 | Me | 2000 | 2004 | 2008 | 2012 | Me |
| S | U | 74% | 71% | 67% | 90% | 76% | 96% | 90% | 96% | 88% | 93% | 88% | 91% | 92% | 92% | 91% |
| | M | 95% | 84% | 98% | 88% | 91% | 84% | 88% | 80% | 76% | 82% | 95% | 89% | 96% | 81% | 90% |
| | O | 96% | 94% | 93% | 94% | 94% | 97% | 97% | 95% | 91% | 95% | 97% | 97% | 96% | 93% | 96% |
| | K | 0.82 | 0.74 | 0.77 | 0.85 | 0.80 | 0.88 | 0.87 | 0.85 | 0.77 | 0.84 | 0.90 | 0.88 | 0.86 | 0.83 | 0.87 |
| C | U | 84% | 83% | 83% | 64% | 79% | 86% | 89% | 93% | 97% | 91% | 86% | 88% | 91% | 90% | 89% |
| | M | 86% | 90% | 90% | 99% | 91% | 84% | 94% | 93% | 85% | 89% | 88% | 94% | 86% | 93% | 90% |
| | O | 97% | 97% | 95% | 85% | 94% | 97% | 99% | 97% | 97% | 98% | 100% | 100% | 99% | 99% | 100% |
| | K | 0.83 | 0.85 | 0.83 | 0.68 | 0.80 | 0.85 | 0.91 | 0.88 | 0.91 | 0.89 | 0.87 | 0.91 | 0.88 | 0.91 | 0.89 |
| H | U | 88% | 88% | 87% | 87% | 88% | 83% | 72% | 62% | 57% | 69% | 89% | 85% | 80% | 83% | 84% |
| | M | 84% | 81% | 81% | 75% | 80% | 93% | 98% | 98% | 97% | 97% | 91% | 89% | 90% | 90% | 90% |
| | O | 99% | 99% | 99% | 99% | 99% | 99% | 99% | 99% | 99% | 99% | 99% | 99% | 99% | 99% | 99% |
| | K | 0.86 | 0.85 | 0.84 | 0.80 | 0.84 | 0.88 | 0.83 | 0.76 | 0.72 | 0.80 | 0.90 | 0.87 | 0.85 | 0.86 | 0.87 |
| MU | | 81% | | | | | 84% | | | | | 88% | | | | |
| MM | | 88% | | | | | 89% | | | | | 90% | | | | |
| MO | | 96% | | | | | 97% | | | | | 98% | | | | |
| MK | | 0.81 | | | | | 0.84 | | | | | 0.88 | | | | |

C, City; Y, Year; Me, Mean; S, Shenyang; C, Changchun; H, Harbin; U, user accuracy; M, Mapping accuracy; O, overall accuracy; K, Kappa coefficients; MU, Mean of user accuracies; MM, Mean of mapping accuracies; MO, Mean of overall accuracies; MK, Mean of Kappa coefficients.

**Table 5. Confusion matrix for the land cover classification.**

| City | 2000 | | 2004 | | 2008 | | 2012 | |
|---|---|---|---|---|---|---|---|---|
| | O | K | O | K | O | K | O | K |
| Shenyang | 97% | 0.94 | 95% | 0.92 | 96% | 0.95 | 94% | 0.90 |
| Changchun | 97% | 0.96 | 96% | 0.90 | 92% | 0.84 | 95% | 0.91 |
| Harbin | 97% | 0.94 | 97% | 0.95 | 97% | 0.96 | 94% | 0.88 |
| Mean | 97% | 0.95 | 96% | 0.92 | 95% | 0.91 | 95% | 0.90 |

O, Overall accuracy; K, Kappa coefficients.

vegetation, water and impervious surfaces; the construction of the ecological environments in Shenyang and Harbin were being strengthened, while Changchun was still focused on the construction of artificial facilities. The pillar industry of provincial capitals in northeastern China is industry. Compact urbanization needs to transfer the industrial facilities in the city center to the periphery, so as to improve the land use efficiency in the main urban area. The paper had shown that the land use efficiency in the main urban areas is improving, and Shenyang and Harbin have advantages in ecological environment. It has been said that further improvement of land use efficiency is very beneficial to the economic development of northeastern China.

Although the method in this paper improves the accuracy of the extraction of urban built-up areas based on remote sensing data, there are still some shortcomings to be studied. First, the analysis of compact urbanization in this paper is limited to the land cover evolution within built-up areas, without considering the impact of other socio-economic factors, and the proposals for the development of compact cities are less well covered. Second, this paper adopted

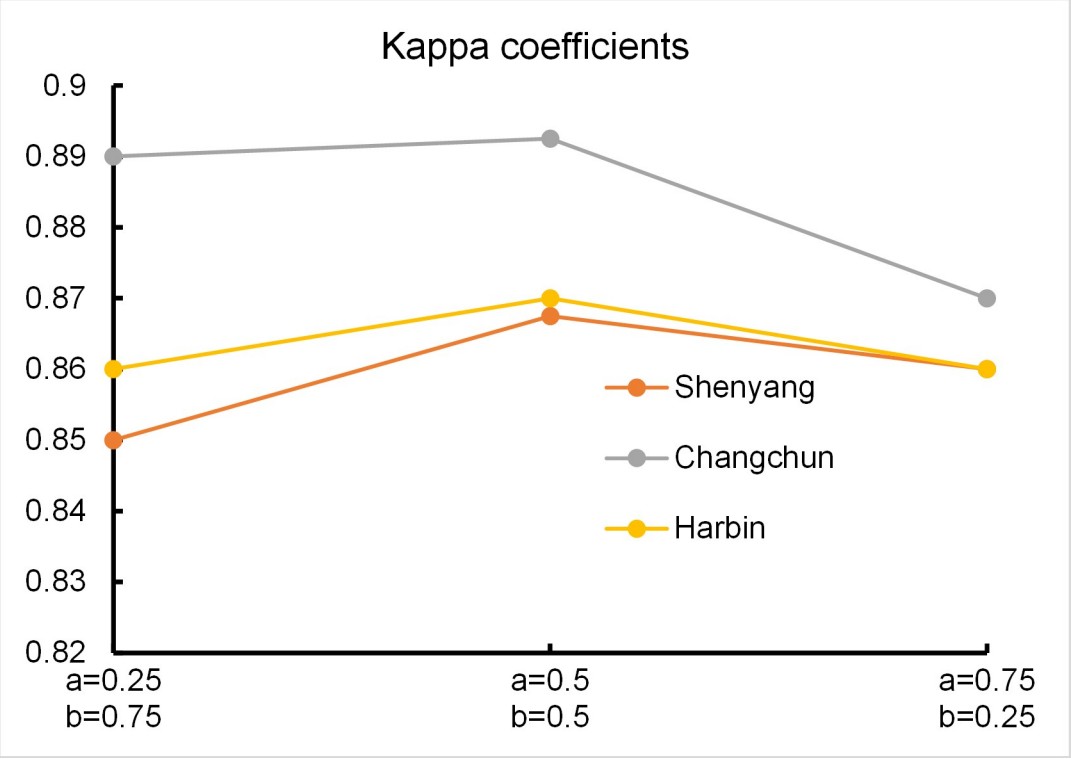

**Fig 13. Boundary pixel fusion weight analysis.**

the DMSP/OLS data as a nighttime light data source; these data are subject to certain constraints due to the acquisition time and spatial resolution, and the subsequent consideration would be to select VIIRS/NPP data, which have a higher spatial resolution. Third, the spatial resolution of the Landsat data still poses a disadvantage, and the selection of GF-2, WordView-2 or other high-resolution remote sensing images could improve the extraction accuracy, but the amount of data operations would increase; reasonable classification methods need further study. In addition, this paper only used landscape indexes to describe the land cover evolution and did not predict evolution trends; in future research, integration with cellular automata (CA), Markov, and other predictive models will be the focus of our approach.

## Acknowledgments

We gratefully thank the data distribution agencies who provided the publicly released data used in this study. The DMSP/OLS data and the Landsat data are obtained from the National Oceanic and Atmospheric Administration (NOAA) and National Aeronautics and Space Administration (NASA), respectively.

## Author Contributions

**Conceptualization:** Zhiwei Xie, Lishuang Sun.

**Data curation:** Jiwei Ping.

**Formal analysis:** Lishuang Sun.

**Methodology:** Yaohui Han.

**Supervision:** Zhiwei Xie, Lishuang Sun.

**Validation:** Zhiwei Xie, Jiwei Ping.

**Writing – original draft:** Yaohui Han.

**Writing – review & editing:** Zhiwei Xie, Yaohui Han.

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
