## [Decision Letter · Decision Letter 0]

30 Jun 2020

PONE-D-20-18472

Analysis of land cover evolution within the built-up areas of provincial capital cities in northeastern China based on nighttime light data and Landsat data

PLOS ONE

Dear Dr. Sun,

Thank you for submitting your manuscript to PLOS ONE. After careful consideration, we feel that it has merit but does not fully meet PLOS ONE’s publication criteria as it currently stands. Therefore, we invite you to submit a revised version of the manuscript that addresses the points raised during the review process.

We look forward to receiving your revised manuscript.

Kind regards,

Bing Xue, Ph.D.

Academic Editor

PLOS ONE

Journal Requirements:

2. Our internal editors have looked over your manuscript and determined that it is within the scope of our Cities as Complex Systems Call for Papers. This collection of papers is headed by a team of Guest Editors for PLOS ONE: Marta Gonzalez (University of California, Berkeley) and Diego Rybski (Potsdam Institute for Climate Impact Research).

The Collection will encompass a diverse and interdisciplinary set of research articles applying the principles of complex systems and networks to problems in urban science.  Additional information can be found on our announcement page: https://collections.plos.org/s/cities.

If you would like your manuscript to be considered for this collection, please let us know in your cover letter and we will ensure that your paper is treated as if you were responding to this call. If you would prefer to remove your manuscript from collection consideration, please specify this in the cover letter.

"This research was funded by Liaoning province education department scientific research project (Grant No. lnqn201917), and Liaoning social science planning fund project (Grant No. L19AJY008)"

4. We note that Figures 4, 6 and 7 in your submission contain map images which may be copyrighted. All PLOS content is published under the Creative Commons Attribution License (CC BY 4.0), which means that the manuscript, images, and Supporting Information files will be freely available online, and any third party is permitted to access, download, copy, distribute, and use these materials in any way, even commercially, with proper attribution. For these reasons, we cannot publish previously copyrighted maps or satellite images created using proprietary data, such as Google software (Google Maps, Street View, and Earth). For more information, see our copyright guidelines: http://journals.plos.org/plosone/s/licenses-and-copyright.

a) You may seek permission from the original copyright holder of Figures 4, 6 and 7 to publish the content specifically under the CC BY 4.0 license. 

5. Please ensure that you refer to Figures 8-10 in your text as, if accepted, production will need this reference to link the reader to the figure.

Additional Editor Comments (if provided):

（1）A professional language editing is required.

Reviewers' comments:

Reviewer's Responses to Questions

**Comments to the Author**

1. Is the manuscript technically sound, and do the data support the conclusions?

Reviewer #1: Yes

Reviewer #2: Yes

2. Has the statistical analysis been performed appropriately and rigorously? 

Reviewer #1: Yes

Reviewer #2: Yes

3. Have the authors made all data underlying the findings in their manuscript fully available?

Reviewer #1: Yes

Reviewer #2: Yes

4. Is the manuscript presented in an intelligible fashion and written in standard English?

Reviewer #1: Yes

Reviewer #2: Yes

5. Review Comments to the Author

Reviewer #1: The research of this paper is valuable. However, the manuscript does not sufficiently explain the scientific significance of the research presented in the manuscript. Some of my observations are given below:

Line 46: If this paper focuses on land cover evolution, should I expect the introduction to start with previous theories on land cover evolution, rather than those build-up area extraction?

The Landsat data built-up area extraction method using maximum likelihood method is a supervised classification method, which requires manual selection of training samples. Why not use the unsupervised classification method, which can improve the automaticity of the method?

All formulas are edited using the Formula Editor (Mathtype software)

Figure 2 and Figure 3 need to be added the results of the remaining two cities.

The background color of Figure 3 (b) needs to be set to white.

The headings of sections 3.2.1 and 3.2.1 shall be consistent with sections 2.2.1 and 2.2.1.

Table 4: What is the meaning of Me.

In the chapter of discussion and conclusion, I suggest to discuss the significance of studying provincial capital cities in northeastern China and the shortcomings of the method in this paper.

Reviewer #2: The article focuses on an important issue, concerning the assessment of land cover evolution, by using remote sensing data. To describe the approach proposed and the model developed, the Authors have presented a case-study related to three major cities in China, analyzing the time interval from 2000 to 2012.

I only have one major issue in this paper which is about the introduction. The introduction started with general background regarding the impact of urbanization and land use and land cover change. Then, in the same paragraph the authors jumped to the data used to extract built-up areas using satellite data and they deeply discussed advanced information such as selection of threshold and segmentation of the nighttime light data. This information is not linked with the topic sentence in the beginning of the paragraph. I suggest to break down this paragraph to be two paragraphs. The first paragraph provides a general information regarding urbanization and land use and land cover change, and the second paragraph deals with the used method to extract built-up information from satellite data. I suggest that the second paragraph discuss the current research status of land cover evolution analysis.

Line 234: What is the basis of the 5 landscape indexes selected and why other indexes were not selected?

Line 262: It is an important process to extract built-up areas using DMSP/OLS or Landsat data alone, and listing corresponding results is helpful to the integrity of the experiment.

Figure 4 needs to have a geographic grid, compass and scale; the lines of indication in Figure 4 (a) and (b) should not cross; the main figure should have a mark.

Lines 117 and 123: Why the selected data are from 2000 to 2012? Reasons should be mentioned.

Figure 6 and Figure 7. The placement of group figures needs to be adjusted.

Figure 11. Text and curve proportion are not in harmony and need to be adjusted.

Take into account the current state of remote sensing data. The reasonableness of the DMSP/OLS data and Landsat data used in this paper requires further explanation, such as why remote sensing data with higher resolution is not used.

6. PLOS authors have the option to publish the peer review history of their article (what does this mean?). If published, this will include your full peer review and any attached files.

Reviewer #1: No

Reviewer #2: No

---

## [Author Response · Author response to Decision Letter 0]

4 Sep 2020

Dear Editors and Reviewers,

Thanks very much for the time and effort you spent in reviewing our manuscript entitled “Analysis of land cover evolution within the built-up areas of provincial capital cities in northeastern China based on nighttime light data and Landsat data” (ID: PONE-D-20-18472). We would like to thank the Editors-in-Chief for giving us a chance to resubmit the paper, and also thank the Associate Editor and the two reviewers for giving us constructive and insightful suggestions. These comments would be very helpful in improving the quality of our paper and provide important guidance to our future research. 

We studied the comments carefully and revised the manuscript in accordance with the comments of the editor and reviewers exactly. Revised portions were marked in red in the revised manuscript. The following is a point-to-point response to the comments of the Associate Editor and the two reviewers.

………………………………………………………………………………………

Academic Editor: 

Comment 1: Response: Thank the Academic Editor very much for his/her constructive comments. We have made changes to the financial disclosure, and updated statement in our cover letter.

Financial Disclosure Statement:

1) Specific grant numbers: 2.

2) Grant numbers awarded to each author: lnqn201917; L19AJY008.

3) The full name of each funder: Education Department Scientific Research Project of Liaoning Province (CN); Social Science Planning Fund Project of Liaoning Province (CN).

4) Initials of authors who received each award: Zhiwei Xie; Zhiwei Xie.

5) Full names of commercial companies that funded the study or authors: Education Department of Liaoning Province; Social Science Planning Fund Office of Liaoning Province.

6) Initials of authors who received salary or other funding from commercial companies: Nobody.

7) URLs to sponsors’ websites: http://jyt.ln.gov.cn/;
http://www.lnsgdb.com.cn/Lnsgdb/publish/html/103/index/index.html.

Comment 2: Guidelines for resubmitting your figure files are available below the reviewer comments at the end of this letter.

Response: Thank the Academic Editor very much for his/her constructive comments. We have resubmitted the figure files as required.

Comment 3: If applicable, we recommend that you deposit your laboratory protocols in protocols.io to enhance the reproducibility of your results. Protocols.io assigns your protocol its own identifier (DOI) so that it can be cited independently in the future.

Response: Thank the Academic Editor very much for his/her constructive comments. The DOI of our Protocols.io is dx.doi.org/10.17504/protocols.io.bifvkbn6.

Journal Requirements：

Comment 1: Please ensure that your manuscript meets PLOS ONE's style requirements, including those for file naming. The PLOS ONE style templates can be found at

https://journals.plos.org/plosone/s/file?id=wjVg/PLOSOne_formatting_sample_main_body.pdf and https://journals.plos.org/plosone/s/file?id=ba62/PLOSOne_formatting_

sample_title_authors_affiliations.pdf

Response: Thank the Academic Editor very much for his/her constructive comments. We have ensured that the manuscript meets the journal style requirements.

Comment 2: Our internal editors have looked over your manuscript and determined that it is within the scope of our Cities as Complex Systems Call for Papers. This collection of papers is headed by a team of Guest Editors for PLOS ONE: Marta Gonzalez (University of California, Berkeley) and Diego Rybski (Potsdam Institute for Climate Impact Research).

The Collection will encompass a diverse and interdisciplinary set of research articles applying the principles of complex systems and networks to problems in urban science. Additional information can be found on our announcement page: https://collections.plos.org/s/cities.

If you would like your manuscript to be considered for this collection, please let us know in your cover letter and we will ensure that your paper is treated as if you were responding to this call. If you would prefer to remove your manuscript from collection consideration, please specify this in the cover letter.

Response: Thanks for the internal editor's approval. If our manuscript is lucky enough to be accepted, we hope it will be published in the main PlOS ONE journal. We think our manuscript does not covers the application of complex systems and networks in urban science. I would prefer to remove our manuscript from collection consideration, and specify this in the cover letter.

Comment 3: Thank you for stating the following in the Acknowledgments Section of your manuscript:

"This research was funded by “Liaoning province education department scientific research project (Grant No. lnqn201917), and Liaoning social science planning fund project (Grant No. L19AJY008)"

Response: Thank you for your reminding, I would remove any funding-related text from the manuscript. Sorry, we didn't find the Financial Disclosure section of the submission form. We had added the Financial Disclosure Statement in the cover letter. Moreover, we entered our funding information in the Funding Information of the submission form, as shown in Response to Reviewers.docx.

Comment 4: We note that Figures 4, 6 and 7 in your submission contain map images which may be copyrighted. All PLOS content is published under the Creative Commons Attribution License (CC BY 4.0), which means that the manuscript, images, and Supporting Information files will be freely available online, and any third party is permitted to access, download, copy, distribute, and use these materials in any way, even commercially, with proper attribution. For these reasons, we cannot publish previously copyrighted maps or satellite images created using proprietary data, such as Google software (Google Maps, Street View, and Earth). For more information, see our copyright guidelines: http://journals.plos.org/plosone/s/licenses-and-copyright.

Response: Thank the Editor very much for his/her constructive comments. Figures 6 and 7 are not satellite images. Figures 6 is results of the extraction of the urban built-up areas based on our method. Figures 7 is land cover class results extracted by us Therefore, these figures do not involve copyright issues. Moreover, the satellite image included in the submitted Figure 4 is Landsat image which is free to download from http://landsat.visibleearth.nasa.gov/.Landsat images are not copyrighted maps.

Comment 5: Please ensure that you refer to Figures 8-10 in your text as, if accepted, production will need this reference to link the reader to the figure.

Response: Yes, we ensure that.

Comment 6: Additional Editor Comments (if provided):（1）A professional language editing is required.

Response: Our manuscript had been professionally edited through American Journal Experts AJE, and the editing certificate is cited in Response to Reviewers.docx

………………………………………………………………………………………

Reviewer #1: 

Comment 1: If this paper focuses on land cover evolution, should I expect the introduction to start with previous theories on land cover evolution, rather than those build-up area extraction?

Response: Thanks a lot for the very helpful comment. We have added previous theories on land cover evolution in the second paragraph of the introduction.

Comment 2: The Landsat data built-up area extraction method using maximum likelihood method is a supervised classification method, which requires manual selection of training samples. Why not use the unsupervised classification method, which can improve the automaticity of the method?

Response: Thanks very much for the valuable comment. The main difference between supervised and unsupervised classifications is whether or not the training sample is selected. Unsupervised classification does not need to select training samples before classification, but it needs to manually judge the attributes of the classes after classification. In addition, unsupervised classification only depends on the distribution of pixel gray value in spectral feature space, which makes it difficult to achieve accurate matching between pixel clusters and ground classes. Because of its simple calculation and strong stability, maximum likelihood method is still widely used in supervised classification algorithm. We added a statement of the reasons for selecting the maximum likelihood taxonomy in line 238.

Comment 3: All formulas are edited using the Formula Editor (Mathtype software)

Response: Thanks a lot for the very insightful comment. We reedited all the formulas using Mathtype Software.

Comment 4: Figure 2 and Figure 3 need to be added the results of the remaining two cities.

Response: We had added the results of the remaining two cities to Figure 2 and Figure 3.

Comment 5: The background color of Figure 3 (b) needs to be set to white.

Response: We had set the background color of Figure 3 (b) to be set to white.

Comment 6: The headings of sections 3.2.1 and 3.2.1 shall be consistent with sections 2.2.1 and 2.2.1.

Response: Thank you very much for your correction. We changed "Nighttime Light Image" to "Nighttime Light Data", and "Landsat image" to "Landsat Data".

Comment 7: Table 4: What is the meaning of Me.

Response: Thank you very much for your correction. To save space, "Me" is an abbreviation of "Mean". We added the above contents in the corresponding position in Table 4.

Comment 8: In the chapter of discussion and conclusion, I suggest to discuss the significance of studying provincial capital cities in northeastern China and the shortcomings of the method in this paper.

Response: Thanks very much for the helpful comment. At the end of the first paragraph of the conclusions section, we had added the discussion about the significance of studying provincial capitals in northeastern China. In addition, the deficiency regarding the analysis of compact urbanization was added in the second paragraph of the same section.

………………………………………………………………………………………

Reviewer #2: 

Comment 1: I only have one major issue in this paper which is about the introduction. The introduction started with general background regarding the impact of urbanization and land use and land cover change. Then, in the same paragraph the authors jumped to the data used to extract built-up areas using satellite data and they deeply discussed advanced information such as selection of threshold and segmentation of the nighttime light data. This information is not linked with the topic sentence in the beginning of the paragraph. I suggest to break down this paragraph to be two paragraphs. The first paragraph provides a general information regarding urbanization and land use and land cover change, and the second paragraph deals with the used method to extract built-up information from satellite data. I suggest that the second paragraph discuss the current research status of land cover evolution analysis.

Response: Thanks very much for the helpful comment. The changes to the manuscript are as follows: (1) we have added previous theories on land cover evolution in the second paragraph of the introduction; (2) the discussion of landscape indices in the introduction section was adjusted to the third paragraph, after the discussion of land cover evolution;(3)The research status of built-up area extraction was adjusted to the fourth paragraph. At the same time, in order to explain the role of the built-up area extraction, the relevant contents of compact urbanization and spread urbanization are added in the fourth paragraph.

Comment 2: What is the basis of the 5 landscape indexes selected and why other indexes were not selected?

Response: Thanks very much for the valuable comment. The principle of selecting landscape indexes is that they can represent the characteristics of ecological state and landscape pattern, and have low redundancy. Landscape index can be divided into three categories: overall trend index, fragmentation index and aggregation index. In combination with the research focus of compact urbanization in this paper, Class area (CA), Percent of landscape (PL), Patch density (PD), Aggregation index (AI) were selected from above categories respectively. We have added the explanation of the reasons for choosing landscape index in line 302 of manuscript.

Comment 3: It is an important process to extract built-up areas using DMSP/OLS or Landsat data alone, and listing corresponding results is helpful to the integrity of the experiment.

Response: We have added the experimental results of built-up area extraction using DMSP/OLS or Landsat data respectively in Figure 6 and 7.

Comment 4: Figure 4 needs to have a geographic grid, compass and scale; the lines of indication in Figure 4 (a) and (b) should not cross; the main figure should have a mark.

Response: Thanks very much for the valuable comment. We had added the geographic grid, compass and scale to Figure 4. The lines of indication in Figure 4 (a) and (b) are redrawn. The main figure has been added the mark (a).

Comment 5: Lines 117 and 123: Why the selected data are from 2000 to 2012? Reasons should be mentioned. 

Response: Thanks very much for the helpful comment. Since DMSP/OLS data were collected from 1992 to 2012, and only the built-up area extraction method using DMSP/OLS was extracted in this paper, the relevant algorithms of VIIRS/NPP were not studied. We had explained this shortcoming in line 501.

Comment 6: The placement of group figures needs to be adjusted.

Response: Thanks very much for the valuable comment. We have adjusted the relative positions between the groups in Figure 6,7 and 8.

Comment 7: Text and curve proportion are not in harmony and need to be adjusted.

Response: We had revised Figure 11 of the original manuscript, and modified the text and curves.

Comment 8: Take into account the current state of remote sensing data. The reasonableness of the DMSP/OLS data and Landsat data used in this paper requires further explanation, such as why remote sensing data with higher resolution is not used. 

Response: DMSP/OLS Data and Landsat data were selected for the following reasons. The area studied in this paper belongs to the geographical objects of medium scale. High resolution remote sensing image can better describe the details, but it can also increase the difficulty of data processing, such as the large amount of data. Therefore, medium and low spatial resolution remote sensing image is more suitable. In addition, most high resolution remote sensing images are non-open source data, so it is difficult for us to obtain relevant data.

………………………………………………………………………………………

In addition, we proofread the manuscript to minimize the typographical, grammatical, and bibliographical errors. Therefore, some sentences and words were rewritten for a native expression, but the meanings did not change. These revisions were not included in the point-to-point response to the comments of the reviewers. However, all revised portions were marked in red in the revised manuscript.

We appreciate for the warm work of the Editors & Reviewers earnestly, and hope that the corrections will meet with approval. Once again, thank you very much for your thoughtful comments and suggestions. Should you have any questions, please contact us without hesitation. Thanks and best regards!

Yours Sincerely,

Lishuang SUN

July 12, 2020

---

## [Editor Report · Decision Letter 1]

7 Sep 2020

Analysis of land cover evolution within the built-up areas of provincial capital cities in northeastern China based on nighttime light data and Landsat data

PONE-D-20-18472R1

Dear Dr. Sun,

We’re pleased to inform you that your manuscript has been judged scientifically suitable for publication and will be formally accepted for publication once it meets all outstanding technical requirements.

Kind regards,

Bing Xue, Ph.D.

Academic Editor

PLOS ONE
---

## [Editor Report · Acceptance letter]

22 Sep 2020

PONE-D-20-18472R1 

Analysis of land cover evolution within the built-up areas of provincial capital cities in northeastern China based on nighttime light data and Landsat data 

Dear Dr. Sun:

I'm pleased to inform you that your manuscript has been deemed suitable for publication in PLOS ONE. Congratulations! Your manuscript is now with our production department. 

Kind regards, 

on behalf of

Professor Bing Xue 

Academic Editor

PLOS ONE